# Efficacy of Interventional Programs in Reducing Acculturative Stress and Enhancing Adjustment of International Students to the New Host Educational Environment: A Systematic Review and Meta-Analysis

**DOI:** 10.3390/ijerph18157765

**Published:** 2021-07-22

**Authors:** Musheer A. Aljaberi, Abdulsamad Alsalahi, Muhamad Hanafiah Juni, Sarah Noman, Ala’a B. Al-Tammemi, Rukman Awang Hamat

**Affiliations:** 1Faculty of Medicine & Health Sciences, Taiz University, Taiz 6803, Yemen; 2Department of Community Health, Faculty of Medicine & Health Sciences, Universiti Putra Malaysia, Serdang 43400, Selangor, Malaysia; hanafiah4660@gmail.com (M.H.J.); saranoman12@gmail.com (S.N.); 3Department of Pharmacology, Faculty of Pharmacy, Sana’a University, Sana’a 1247, Yemen; ahmedsamad28@yahoo.com; 4Department of Epidemiology and Global Health, Faculty of Medicine, Umeå University, 90187 Umeå, Sweden; alaa.tammemi@med.unideb.hu; 5Doctoral School of Health Sciences, University of Debrecen, 4032 Debrecen, Hungary; 6Department of Family and Occupational Medicine, Faculty of Medicine, University of Debrecen, 4032 Debrecen, Hungary; 7Department of Medical Microbiology, Faculty of Medicine & Health Sciences, Universiti Putra Malaysia, Serdang 43400, Selangor, Malaysia

**Keywords:** acculturative stress, adjustment, international students, intervention, systematic review, meta-analysis

## Abstract

This review aimed to systematically outline and meta-analyze the efficacy of psychoeducational, cultural orientation, socio-cultural, and peer-pairing programs in reducing acculturative stress and enhancing adjustment among international students worldwide. The consulted databases were PubMed, Scopus, Web of Science, ScienceDirect, EBSCO, and ProQuest. Eligibility criteria allowed the inclusion of randomized controlled trials (RCTs) and quasi-experimental trials without applying language, country, publication type or time restrictions. The quality of the eligible studies was appraised by the RoB2 tool of Cochrane for RCTs and JBI critical appraisal tools for quasi-experimental trials. Data items were collected based on PICO acronym by two investigators and reviewed for accuracy by a third one. The evidence was narratively synthesized and validated by proceeding with a random model meta-analysis using Cochrane RevMan software(Version 5.4). The quality of the pooled evidence from meta-analysis was assessed using the tool of GRADE. Out of 29,975 retrieved records, 14 studies (six RCTs and eight quasi-experimental trials) were included. The psychoeducational program significantly reduced acculturative stress and enhanced adjustment. In contrast, cultural orientation and peer-pairing programs significantly enhanced adjustment, but could not reduce acculturative stress. In meta-analysis, acculturative stress was significantly reduced in the psychoeducational intervention versus controls [overall pooled size effect = −3.89 (95% CI: −5.42, −2.53) at *p* < 0.001]. Similarly, adjustment was significantly enhanced in the psychoeducation and socio-cultural interventions versus control [overall pooled size effect = 3.10 (95% CI: 2.35, 3.85) at *p* < 0.001]. In conclusion, the psychoeducational program demonstrated superior efficacy in reducing acculturative stress and enhancing adjustment compared to the other interventional programs. However, socio-cultural programs have still been effective in enhancing adjustment. This systematic review is registered in PROSPERO (CRD42018104211).

## 1. Introduction

International students are described as ‘student sojourners’ who move to other countries to pursue their higher education within a particular period [1,2]. The number of international students joining higher educational institutions outside their home countries has overgrown [3,4]. As revealed by the United Nations Educational, Scientific and Cultural Organization (UNESCO) (2009), there were almost 2.5 million international students joining universities worldwide [3,5]. Interestingly, there was a rise in the enrollment of international students worldwide from 2 million in 1999 to 5 million in 2016, which has been estimated to increase exponentially, predicted to rise to fifteen million in the year 2025 [2]. Recently, it has been reported that at least 5.3 million international students are studying abroad, and 43.8% of them are in the USA, the UK, Australia, Germany, and France [6].

International students abroad have become the target of media and research, especially social science research [7]. Regardless of the diversity of the cultural, religious, political, and social features of international students studied abroad, their residence in the host country is temporary to achieve their goals and return to their original countries [8]. However, one of the most serious challenges that international students encounter is the failure to adjust to the host country [9,10,11,12,13,14,15,16,17]. This results in deteriorated health conditions and severe health problems [18,19,20,21], alcohol consumption [21,22], feeling of anxiety [23,24], stress, depression, and physical illness [2,21,25,26,27,28,29,30,31,32,33], which negatively affect students’ academic achievements and lead to their dropout from universities [34,35,36]. In other words, international students experienced acculturative stress, which refers to the various psychological and behavioral stressors that afflict an individual due to being exposed to another culture and society [9,37], which is recognized as a serious and critical issue experienced by international students [2,9,17,38,39].

International students develop acculturative stress and depression in their intercultural transition [2]. On the other hand, international students with inadequate internal and external resources are more likely to experience an increased level of acculturative stress [40]. Gebregergis, Huang and Hong [2] reported that 506 international university students from seven Chinese universities experienced a higher level of acculturative stress with more significant depressive symptoms. Similarly, and Koo, et al. [41] reported a higher level of acculturative stress among first-year international students enrolling in U.S. higher education. Likewise, and Musheer, Juni, Shahar and Ismail [17] reported that 78.5% out of 522 new postgraduate international students joining Malaysian Public Universities experienced a moderate level of acculturative stress, and 12.1% experienced a high level of acculturative stress, while 4.9% of them experienced a low level of acculturative stress. Consequently, 42.1% of the overall number of the participant’s students they have intention to dropout from the university.

Reviewing the literature indicated that several studies had been conducted to evaluate the efficacy of different interventions in reducing international students’ acculturative stress and enhancing their adjustment to the new host environment [42,43,44,45,46,47,48]. Examples of these interventions are psychoeducational, cultural orientation, socio-cultural, and peer-pairing programs. However, the efficacy of such interventional programs in reducing international students’ acculturative stress has still been conflicting and not yet been systemically reviewed. Accordingly, this systematic review and meta-analysis formulated a research question of “In randomized controlled trials and quasi-experimental trials, do the psychoeducational, cultural orientation, socio-cultural, and peer-pairing programs reduce the acculturative stress of international students and enhance their adjustment to the new host environment as compared to untreated or placebo-receiving international students?”. For answering this research question, this review systematically outlined and meta-analyzed evidence of the efficacy of the psychoeducational, cultural orientation, socio-cultural, and peer-pairing programs to reduce acculturative stress among international students and enhance their adjustment to the new host environment worldwide.

## 2. Materials and Methods

Performing and reporting the current systematic review adhered to the 2020-updated PRISMA checklist to achieve reproducibility of the methodology and findings to enhance the reliability of this systematic review and meta-analysis [49]. The prospective protocol for conducting this systematic review is registered in the PROSPERO Platform (CRD42018104211). Since this systematic review relied on published studies, the informed consent of participants or approval of the Institutional Review Board (I.R.B.) was not required. No amendments have been made to the prospective protocol of this systematic review.

### 2.1. Search Strategy

#### 2.1.1. Keywords

The keywords syntax to retrieve the records from online databases were constructed to include intervention and a primary (acculturative stress) or secondary outcome (adjustment). The terms of intervention and students were intentionally applied to achieve a comprehensive retrieval of records from databases. The different spelling and synonyms were considered upon constructing the syntax of keywords to optimize the keywords. Accordingly, the following keywords with Boolean operators were used including ((“students”) and (“intervention” or “program” or “management” or “therapy” or “training”) and (“culture shock” or “acculturative stress” or “acculturation stress”)) and ((“students”) and (“intervention” or “program” or “management” or “therapy” or “training”) and (“adapting” or “coping” or “adjustment”)). Additionally, the following keyword was also applied ((“students”) and (“intervention” or “program” or “management” or “therapy” or “training”) and (“withdrawal” or “dropout” or “retreat”)).

#### 2.1.2. Databases and Information Sources

Four online databases were used to retrieve the pertinent records, including PubMed, Scopus, Web of Science, ScienceDirect, and EBSCO (CINHAL Complete and Psychology and Behavioral Sciences). The same keywords were applied in all the databases to index the pertinent records by title, abstract, and keywords, except PubMed. The title and the abstract were used to index the documents.

Additional sources such as Google Scholar, bibliographies, libraries, and the feedback from authors were considered to retrieve extra records.

For grey literature, several databases, including ProQuest and MedNar, were searched to retrieve unpublished relevant interventional studies to avoid publication bias.

#### 2.1.3. Filters

In this review, no filters (by language, country, or type of records documents) were applied. The time frame included any published papers at any time in previous years until 4 November 2019.

#### 2.1.4. Teamwork

The search strategy was implemented by two independent researchers (M.A.A. and A.A.) were involved. In cases of discrepancy, a discussion between the two independent researchers and/or participation of a third researcher (R.A.H.) was involved.

### 2.2. Study Selection

The total number of identified records from each retrieved database or other sources were recorded. Then, duplicates were removed.

#### 2.2.1. Primary Selection

It was planned to restrict the selection of records to research articles and theses through screening them by titles, abstracts. Accordingly, books, conferences, and different elements (e.g., indices, glossaries, lists, and bibliographies) were removed. Then, the remaining records were screened by titles and abstracts for a related content of the primary (acculturative stress) and secondary (adjustment) outcomes. At the same time, unrelated records were excluded (Figure 1).

#### 2.2.2. Secondary Selection

The related records underwent a secondary selection to select the relevant records by screening full text according to the pre-specified inclusion and exclusion criteria of eligibility (Table 1).

#### 2.2.3. Teamwork

The primary and secondary selection of records was achieved by two independent researchers (M.A.A. and A.A.). In cases of discrepancy, a discussion between the two independent researchers and/or participation of a third researcher (R.A.H.) was involved.

### 2.3. Quality Assessment at the Level of Eligible Studies

#### 2.3.1. Randomized Controlled Trials

The RoB2 tool of the RevMan software program (Version 5.4; Cochrane Collaboration, Oxford, UK) was applied to assess the risks of bias at the level of the RCT [50,51,52]. The domains of the RoB2 tool were selection bias (two domains, including random generation sequence and allocation concealment), performance bias, detection bias, attrition bias, reporting bias, and other sources of bias. The appraisal of these domains in each RCT was assessed as either a high, low, or unclear risk of bias (Table A1 (Appendix A)).

The decision to exclude an RCT was made if three events of a high risk of bias were identified [50,51,52,53]. Otherwise, each RCT fulfilled the criteria of low risk (robust internal validity). The decision was made to include the RCT to extract the data to synthesize qualitative and quantitative literature reviews.

The assessment of the risk of vias was implemented by two independent researchers (M.A.A. and A.A.). In cases of discrepancy, a discussion between the two independent researchers and/or participation of a third researcher (R.A.H.) was involved.

#### 2.3.2. Non-Randomized Intervention Trials (Quasi-Experimental Trials)

The assessment of the risk of bias at the level of the quasi-experimental trials was performed by using JBI critical appraisal tools [54]. The answer to the signaling questions was defined either as “yes” (low risk of bias), “no” (high risk of bias), inapplicable assessment, or unclear (inability to be assessed because of the absence of direct or indirect evidence) (Table A2 (Appendix B)).

Similar to the RCTs, after assessing the risks of bias within each quasi-experimental trial, the judgment of exclusion criteria was based on the presence of four domains scoring a high risk of bias [53]. Moreover, the overall assessment of each type of bias across the included studies was considered.

The assessment of the risk of bias was implemented by two independent researchers (M.A.A. and A.A.). In cases of discrepancy, a discussion between the two independent researchers and/or participation of a third researcher (R.A.H.) was involved.

### 2.4. Data Collection

#### 2.4.1. Strategy

The data were extracted from tables and/or texts of the low-biased studies and summarized precisely in a standard excel spreadsheet. In case of missing data, it was planned to contact the authors (Table A3 (Appendix C)).

#### 2.4.2. Data Items

Study design

First author, year of publication, type of study design, and settings.

Population (participant’s)

Sample size, nationality, and gender.

Intervention

Type of intervention, duration of intervention, number of sessions, frequency of sessions, and number of participants.

Comparators

Placebo, untreated (waiting list) or standard therapy; number of participants.

Primary outcomes

Acculturative stress was recruited as a primary outcome, which is defined as the individual’s psychological and physical tension when attempting to adjust to a new culture measured by reliable measurements at the post-intervention time points and estimated with mean ± standard error or standard deviation.

Secondary outcomes

Adjustment to the new environment was recruited as a secondary outcome, which is defined as either psychological or socio-cultural adjustment evidenced with feelings of wellbeing, satisfaction, or ability to adjust to a new environment measured by reliable measurements at the post-intervention time points and estimated with mean ± standard error or standard deviation.

The process of data extraction and collection involved two independent researchers (M.A.A. and A.A.). A third investigator was involved to ensure the accuracy of the extracted data (R.A.H.).

### 2.5. Strategy for Data Synthesis

#### 2.5.1. Qualitative Literature Review

The evidence of the efficacy of all interventional programs on either acculturative stress, adjustment to the new host environment or both acculturative stress and adjustment to the new host environment were narratively synthesized, separately. However, the conclusion from the narrative review was planned to be validated by conducting a subsequent meta-analysis.

#### 2.5.2. Quantitative Literature Review (Meta-Analysis)

The quantitative data of acculturative stress and adjustment outcomes in the intervention and control groups estimated as mean ± standard deviation was enrolled in the meta-analysis using The ReveMan Software of Cochrane. The effect size (ES) index was computed as the mean difference between the intervention and control groups at 95% confidence intervals of the weighted average effect size (positive or negative). The I^2^ was used to indicate heterogeneity of the pooled mean difference effect. Sensitivity analyses were applied through repeating meta-analysis and using the size effect for each subset [55]. Finally, the random-effect model of the meta-analysis was applied to elucidate the effect size, assuming that the included trials measured the interventions with higher heterogeneity [56].

### 2.6. Quality of the Pooled Evidence from a Meta-Analysis

The quality of pooled evidence from meta-analysis after applying the sensitivity test was assessed using the tool of GRADE (Grading of Recommendations, Assessment, Development, and Evaluations) [57,58]. The overall grade estimate could be low, moderate, or high after grading the domains of study design (RCT or observational), indirectness, risk of bias, imprecision, inconsistency, and publication bias. Evidence was considered to have a high-quality grade if the certainty in the pooled evidence from each meta-analyzed subset was high.

The quality of evidence from meta-analysis was upgraded because the evidence is obtained from RCTs rather than quasi-experimental design. Additionally, the quality of evidence is obtained from low risk of bias studies. Moreover, the quality of evidence is upgraded when achieving the conditions of consistency (overlapped confidence intervals and I^2^ < 50% at *p* > 0.05), directness (measured outcome related to the participants of interest, and the interventions and controls were compared head-to-head), precise (narrow confidence intervals around the effect size estimate were narrow) and absence of publication bias.

## 3. Results

### 3.1. Selected Studies

A total of 29,975 records were retrieved from the databases (*n* = 24,860) and other sources (*n* = 5115). According to the inclusion and exclusion criteria, 20 records (research article = 14 and theses = 6) were eligible [42,43,44,45,46,47,48,59,60,61,62,63,64,65,66,67,68,69,70,71] for further selection process (Figure 1).

### 3.2. Quality of the Eligible Studies

Upon applying the risk of bias assessment tools, 14 studies out of the 20 eligible studies met the criteria of low risk of bias from which the data were extracted [42,43,44,46,47,48,59,60,61,62,65,67,69,71]. The 14 low-biased included studies were six randomized interventional studies [16,47,48,59,61,62] and eight non-randomized interventional studies (Quasi-experimental design) [42,43,44,46,60,67,69,71]. Figure 2a shows that the risk of bias across the included RCTs (*n* = 6) was generally low due to the implementation of adequate randomized assignment, unselective reporting of the outcomes, adequate addressing of missing data and withdrawals, and adherence to the prospective protocols of the RCTs.

Table 2 shows that the risk of bias across quasi-experimental trials was high regarding the inadequate application of multiple measurements for the outcomes at several time points (pre-and post-measurement) and the incomplete follow-up for the participants from the time of implementing the intervention to the time of cut-off date. However, the risk of bias was low regarding the causal relationship between independent and dependent variables (causal–effect relationship), balancing the characteristics of comparisons, applying similar treatment/care for comparisons, recruiting independent control group, using similar outcomes measurement for compressions, using reliable outcome measurement and applying appropriate statistical methods.

Table 2 also shows the results of the assessment of the risks of bias within each included study. All the included studies met the criteria of low risk of bias according to the overall judgment criteria of high-risk, indicating that these studies were eligible to provide highly reliable and valid data about the measured primary (acculturative stress) and secondary (adjustment) outcomes.

### 3.3. Characteristics the Included Studies

All the included trials were interventional (*n* = 14) with high quality (low risk of bias). It could be noticed that the implemented interventions (psychological, psychoeducational, cultural orientation, socio-cultural, and peer-pairing) in the included studies shared some overlapping components or features. Therefore, this systematic review analyzed and appraised the effect of those interventions on the level of acculturative stress of international students or their adjustment to the new environment. The 14-included studies were six RCTs and eight quasi-experimental designs, which covered a timeframe from 1988 to 2019 and were conducted in six countries, including Australia, Canada, Costa Rica, Malaysia, Turkey, and the United States of America (USA). The included studies enrolled 824 international students who met the criteria of suffering from acculturative stress and/or inability to adjust to the new environment. These participated students came from at least 59 nationalities worldwide (Figure 3). Regarding gender, 409 males and 368 females were involved in all the included trials.

### 3.4. Narrative Systematic Literature

#### 3.4.1. Effects on Acculturative Stress of the International Students

Eight studies evaluated the efficacy of different interventional programs (cultural orientation, psycho-educational, psychological, and peer-pairing) to reduce international students’ acculturative stress [47,48,59,60,61,62,67,71]. Four studies showed that psychoeducational and psychological interventional programs effectively reduced acculturative stress [47,60,61,62]. Conversely, four studies reported that the implementation of cultural orientation programs [42,71], psychological programs [48], and peer-pairing programs [68] could not significantly reduce acculturative stress (Table A3).

#### 3.4.2. Effects on Adjustment of International Students to the New Environment

A total of nine studies evaluated the efficacy of different interventional programs (psychoeducational, cultural orientation, peer-pairing, and socio-cultural programs) on the adjustment of international students to the new host environment [42,43,44,46,62,65,67,69,71]. Two trials implemented social-cultural interventional programs, which could not significantly enhance the adjustment [44,65]. Conversely, the other seven trials implemented psycho-educational programs [43,62], cultural orientation programs [46,71], and peer-pairing programs [42,67,69], which significantly enhanced the adjustment. However, each program used a different scale to measure the outcomes (Table A3).

#### 3.4.3. Effects on Acculturative Stress and Adjustment of International Students

Three trials concurrently evaluated the effect of different interventional programs (cultural orientation, peer-pairing, and psychoeducational programs) on acculturative stress and adjustment of the international students [62,67,71]. The results showed that psychoeducational programs could significantly reduce acculturative stress and enhance their adjustment simultaneously [62]. In contrast, the cultural orientation [71] and peer-pairing programs [67] significantly improved the adjustment. However, both programs did not reduce acculturative stress (Table A3).

### 3.5. Quantitative Literature (Meta-Analysis)

Nine trials were enrolled in the meta-analysis [42,44,47,48,60,61,62,65,67]. Six out of them were for the data of acculturative stress, including four RCTs [47,48,61,62] and two quasi-interventional trials [60,67]. For the data of adjustment, five trials were enrolled, including two RCTs [62,65] and three quasi-interventional trials [42,44,67].

The RCT by Tavakoli et al. (2009) [48] was subdivided into three studies (Tavakoli et al., 2009a, 2009b and 2009c) because it contains three different interventions (assertive training, expressive writing, and a combination of the two methods) versus the same control group (Table A4 (Appendix D)). For each outcome, the data were meta-analyzed before and after the application of the sensitivity test each time.

#### 3.5.1. Acculturative Stress of the International Students

Before application of the sensitivity test

The data of eight studies were enrolled in the meta-analysis. The overall pooled effect size of acculturative stress in the intervention group was not significantly reduced versus control [mean difference: −0.36 (95% CI: −0.72, 0.00) at *p* = 0.05] with a highly significant difference in the heterogeneity (I^2^ = 81% at *p* < 0.00001) (Figure 4a).

After application of the sensitivity test

The data of three RCTs evaluating the efficacy of psychoeducational program on acculturative stress were enrolled in the meta-analysis. The overall pooled effect size of acculturative stress in the psychoeducational intervention group was significantly reduced versus control [mean difference: −3.89 (95% CI: −5.42, −2.53) at *p* < 0.00001] without a significant difference in heterogeneity (I^2^ = 0% at *p* = 0.69) (Figure 4b).

#### 3.5.2. Adjustment of the International Students

Before application of the sensitivity test

The data of five studies were enrolled in the meta-analysis. The overall pooled effect size of adjustment in the intervention groups was significantly enhanced versus control [mean difference: 2.46 (95% CI: −0.72, 4.20) at *p* = 0.006]; however, the heterogeneity was significantly high (I^2^ = 98% at *p* < 0.00001) (Figure 5a).

After application of the sensitivity test

The data of two RCTs were enrolled in the meta-analysis. The overall pooled effect size of adjustment in the psychoeducational and socio-cultural intervention groups was significantly enhanced versus control [mean difference: 3.10 (95% CI: 2.35, 3.85) at *p* < 0.00001] without a significant difference in the heterogeneity (I^2^ = 2% at *p* = 0.31) (Figure 5b).

#### 3.5.3. Assessment of the Quality of Evidence Pooled from the Meta-Analysis

Before applying the sensitivity test, publication bias was detected in the pooled evidence of acculturative stress (Figure 6a) and adjustment (Figure 6c) due to the asymmetry of studies in the funnel plots. However, after applying the sensitivity test, the publication bias in the pooled evidence of acculturative stress (Figure 6b) and adjustment (Figure 6d) was not detected due to the symmetry in the funnel plots.

After applying the sensitivity test, the evidence of acculturative stress and adjustment of the studies enrolled in this meta-analysis were up-graded because the pooled evidence came from RCTs without a serious risk of bias. In addition, the pooled evidence had high quality due to the publication bias was not detected without serious risks of inconsistency, indirectness, and imprecision (Table 3).

## 4. Discussion

### 4.1. Summary of the Evidence

The findings of the narrative systematic literature indicated that psychoeducational programs reduced acculturative stress of international students [47,60,61,62] and enhanced their adjustment to the new host environments [66]. The former findings were validated by conducting a meta-analysis. The overall pooled effect size of the psychoeducational intervention significantly reduced acculturative stress of international students compared to the control group [47,61,62]. Similarly, the overall pooled effect size of the psychoeducational intervention significantly enhanced the adjustment of international students to their new host environments compared to the control group [62,65]. Accordingly, the evidence of the narrative systematic literature and meta-analysis indicated that the psychoeducational programs are efficacious in reducing acculturative stress of the international students and enhancing their adjustment to the new host environments. Meanwhile, the evidence from the narrative systematic literature indicated that the cultural orientation [46,71] and peer-pairing programs [42,67,69] significantly enhanced the adjustment of international students to the new host environments, but failed to reduce their acculturative stress. These findings were validated by conducting the meta-analysis in which the overall pooled effect size of the socio-cultural interventional program was the only program that significantly enhanced the adjustment as compared to the control group [62,65].

The most critical challenge that encounters international students is the adjustment to the new host environments, which might contribute to their susceptibility to intense emotional experiences and other mental health issues [43]. One of the most manifestations of such psychological problems is acculturative stress [18,19,20,21], associated with feelings of stress, anxiety, depression, and physical illness [2,23,24,25,26,27,28]. The acculturative stress is developed because the international students encounter considerable difficulty in making new adjustments because of the lack of social support, communication problems, and homesickness [72]. For such reasons, the psychoeducational programs were superior to the other programs in reducing acculturative stress and enhancing the adjustment of the international students to the new host environments. Perhaps the efficacy of the psychoeducational programs could be because this program targeted all the aspects of acculturative stress, including the psychological sources. At the same time, the other interventional programs focused only on the socio-cultural aspects while ignoring the psychological source of acculturative stress.

### 4.2. Limitations of Current Work

Regarding limitations of the included trials in this systematic review and meta-analysis, the included studies lack strict study designs and adequate sample size to optimize the power of the statistical analysis for detecting and measuring the outcomes at different timepoints during the time course of the intervention, which could affect the maturity of the measured outcomes (i.e., the outcomes did not reach a significant level). Additionally, the included studies disclose that the instruments and measurement of the acculturative stress and adjustment are not uniform, which could affect the consistency of the measured outcomes. Regarding the limitation of this systematic review, this review included the trials that published in the English language. Besides, there has been a limited number of studies in the literature that focusing on evaluating the efficacy of the international programs to reduce acculturative stress and enhance adjustment of international students to the new host environments.

### 4.3. Implications of the Current Work

Although the results of the narrative systematic review were expressive in demonstrating that the evidence of the randomized controlled trials and quasi experimental trials were informative, the meta-analysis validated only the evidence that were extracted from the randomized controlled trials, of which the quality was upgraded to high certainty making them reliable to be relied on by the policymakers and interested researchers.

For implications of the results of this systematic review and meta-analysis for practice, policy, and future researches, a specific notice that most universities worldwide are interested in attracting international students by providing several advantages such as international ranking, advanced educational techniques, and accommodations. Unfortunately, most of those universities drop from their account the acculturation of international students with local societies, which will help international students avoid acculturative stress and dropout. This systematic review recommends a comprehensive intervention study with a standardized design of randomized controlled trials and adequate sample size incorporating cognitive, behavioral, psychological, social, and affective elements. This is essential for facilitating the cross-cultural adaptation of international students and accommodating their unique needs in acculturation, and reducing their acculturative stress and their dropout rates from the universities. In addition, this systematic review and meta-analysis imply the need for developing comprehensive instruments to fulfil the covariates such as gender, ethnicity, age, education level, and cultural backgrounds of the international students, which in our opinion are expected to make the internal validity and external validity (generalizability or applicability) of the conducted trials more robust.

## 5. Conclusions

This systematic review and meta-analysis make important contributions to draw an informed conclusion about the most effective interventional programs to reduce international students’ acculturative stress and enhance their adjustment to the host environment. Consequently, evidence from the qualitative and quantitative indicated that the psychoeducational programs were the only efficacious interventional program to reduce international students’ acculturative stress. In contrast, the psychoeducational and socio-cultural interventional programs were efficacious in enhancing the adjustment of the international students to the new host environment. Such findings could indicate that psychoeducational programs are superior to the other interventional programs in reducing international students’ acculturative stress and enhancing their adjustment to the new host environment. However, this systematic review and meta-analysis recommend a comprehensive psychoeducational interventional program with a large sample-sized standardized study design that incorporates cognitive, behavioral, psychological, social, and affective elements with the utilization of relevant, valid, and reliable measurement to evaluate the acculturation outcomes of international students.

## Figures and Tables

**Figure 1 ijerph-18-07765-f001:**
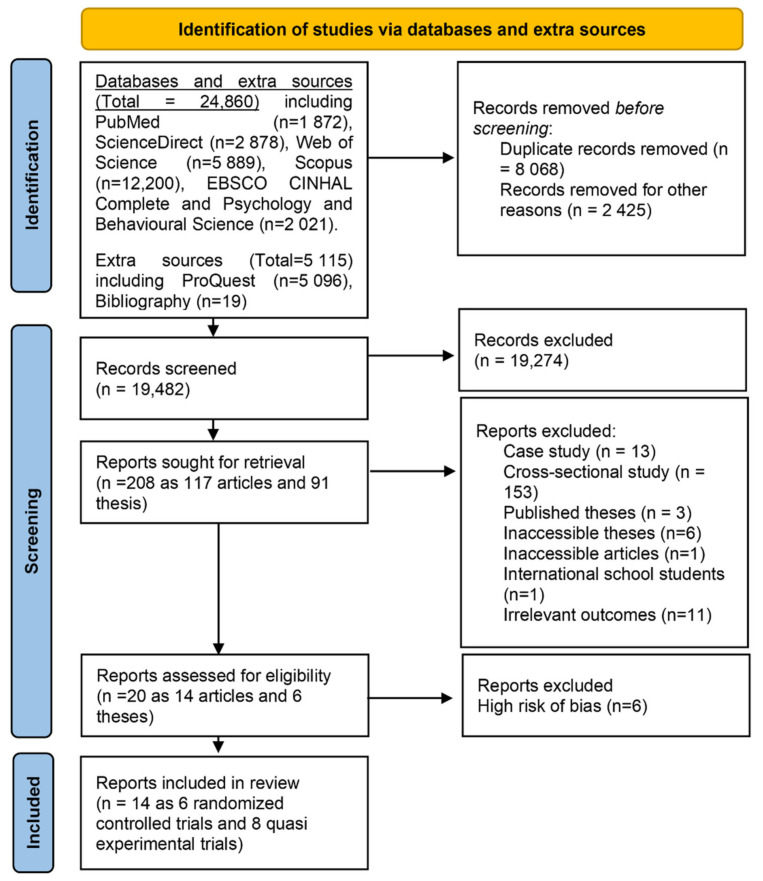
Flowchart of the methodology of the current study according to PRISMA.

**Figure 2 ijerph-18-07765-f002:**
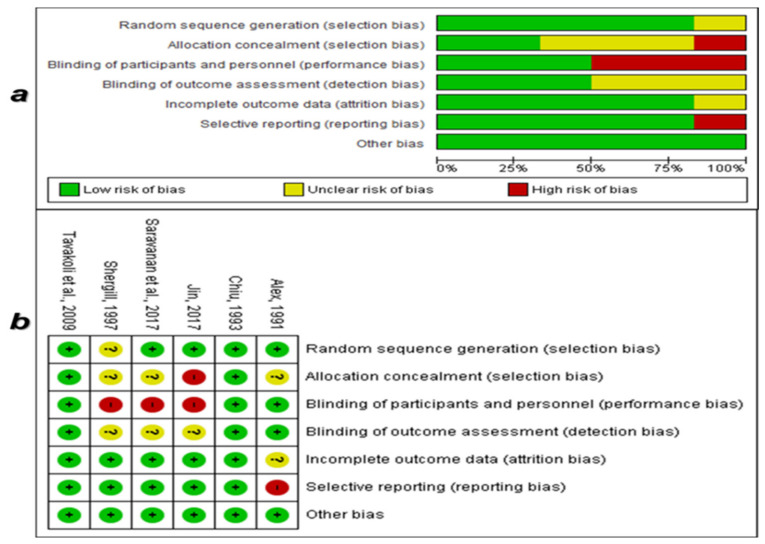
Quality assessment of the risk of bias in the randomized interventional studies. (**a**) Risk of bias across studies, (**b**) risk of bias within each study.

**Figure 3 ijerph-18-07765-f003:**
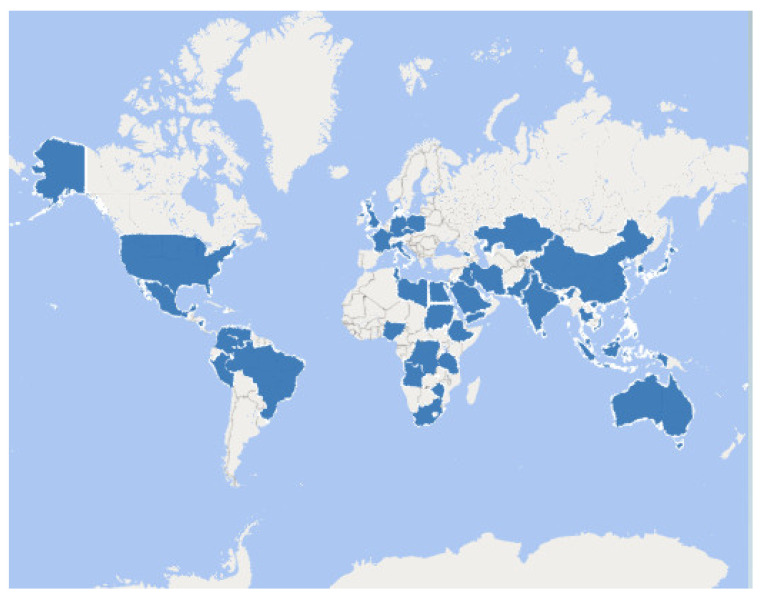
Nationalities of the participating international students in the included studies. The dark blue color indicates the countries of participating international students.

**Figure 4 ijerph-18-07765-f004:**
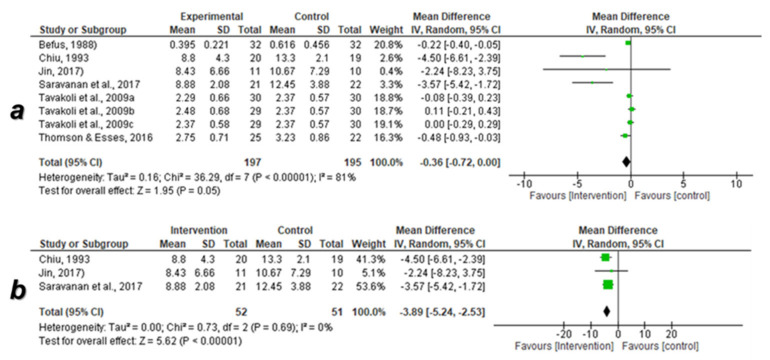
Forest plot of the meta-analysis of the data of acculturative stress. (**a**) Before application of sensitivity test, (**b**) after application of sensitivity test, I^2^: heterogeneity, CI: confidence interval, SD: standard deviation, P: *p*-value. Black diamond denotes the overall effect size, green color denotes the central point of the confidence interval. The RCT by Tavakoli et al. (2009) [48] was subdivided into three studies (Tavakoli et al., 2009a, 2009b and 2009c) because it contains three different interventions (assertive training, expressive writing, and a combination of the two methods) versus the same con-troll group.

**Figure 5 ijerph-18-07765-f005:**
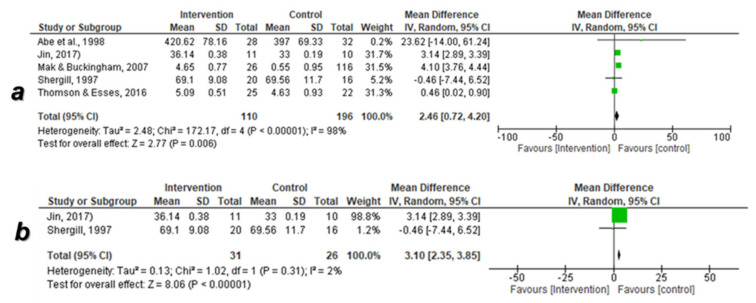
Forest plot of the meta-analysis of the data of adjustment. (**a**) Before application of sensitivity test, (**b**) after application of sensitivity test. I^2^: heterogeneity, CI: confidence interval, SD: standard deviation, P: *p*-value. Black diamond denotes the overall effect size, green color denotes the central point of the confidence interval. Black diamond denotes the overall effect size, green color denotes the central point of the confidence interval. The RCT by Tavakoli et al. (2009) [48] was subdivided into three studies (Tavakoli et al., 2009a, 2009b and 2009c) because it contains three different interventions (assertive training, expressive writing, and a combination of the two methods) versus the same con-troll group.

**Figure 6 ijerph-18-07765-f006:**
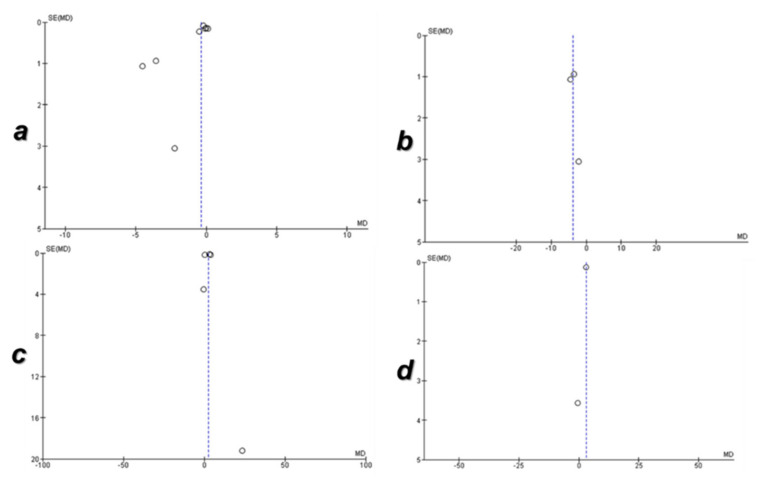
Funnel plots of the publication bias of pooled evidence from a meta-analysis. (**a**) Funnel plot of evidence of acculturative stress before application of the sensitivity test, (**b**) funnel plot of evidence of acculturative stress after application of the sensitivity test, (**c**) funnel plot of evidence of adjustment before application of the sensitivity test, (**d**) funnel plot of evidence of adjustment after application of the sensitivity test.

**Table 1 ijerph-18-07765-t001:** The inclusion and exclusion criteria according to the statement of PICOS *.

		Included	Excluded
Conditions and domains		• Acculturative stress and adjustment to the new host environment among international students worldwide	• Irrelevant to acculturative stress and adjustment among international students
Population		• Worldwide enrolled international universities or college students study out of their original countries (any gender, ethnicity, education level, any nationality)	• Online international students • Local students/students at schools • Refugees/Immigrants for purposes other than study
Interventions		• Any interventional programs (psycho-educational, cultural orientation, socio-cultural, and peer-pairing programs) were implemented to minimize acculturative stress and enhance their adjustment to new host culture	• Medicated intervention
Comparators		• Comparator(s) not exposed to any intervention • Baseline assessment as a comparator	• None
Study Design		• Randomized controlled and quasi-experimental trials that have been conducted to reduce acculturative stress or enhance the adjustment among international students using psycho-educational, cultural orientation, socio-cultural, and peer-pairing programs	• Non-interventional trials
Outcomes	Primary	• Acculturative stress	• Irrelevant
	Secondary	• Adjustment	• Irrelevant
Others (article type)		• Published research articles in peer-reviewed journals • Unpublished theses	• Inaccessible research articles • Published theses

* PICOS: acronym of Population, Intervention, Comparator, Outcomes, and Study design.

**Table 2 ijerph-18-07765-t002:** Quality assessment of the risk of bias in the non-randomized interventional studies (Quasi design).

Study ID	Causal Relationships	Balanced Characteristics for Comparisons	Similar Treatment/Care for Comparisons	Independent Control Group	Pre- and Post-Measurement (Multiple Measurements)	Complete Follow-Up for Comparisons	Similar Outcomes Measurement for Comparisons	Reliable Outcome Measurement	Appropriate Statistical Methods
[43]	Yes	Yes	Yes	Yes	No	No	Yes	Yes	Unclear
[44]	Yes	Nuclear	Yes	Yes	Yes	Yes	Yes	Yes	Yes
[61]	Yes	Yes	Yes	Yes	No	No	Yes	Yes	Yes
[45]	Yes	Yes	Yes	Yes	Yes	No	Yes	Yes	Yes
[47]	Yes	Yes	Yes	Yes	No	Yes	Yes	Yes	Yes
[68]	Yes	Unclear	Yes	Yes	Yes	No	Yes	Yes	Yes
[70]	Yes	Yes	Yes	Yes	Yes	No	Yes	Yes	Unclear
[72]	Yes	Yes	Yes	Yes	Yes	No	Yes	Yes	Yes

Yes: Low risk of bias, No: High risk of bias, Inapplicable: High risk of bias, Unclear: Could not be evaluated well.

**Table 3 ijerph-18-07765-t003:** Assessment of pooled evidence from meta-analysis after application of sensitivity test.

Certainty Assessment							Certainty
№ of Studies	Study Design	Risk of Bias	Inconsistency	Indirectness	Imprecision	Other Considerations *	
Acculturative stress							
3	Randomized trials	not serious	not serious	not serious	not serious	none	HIGH
Adjustment							
2	Randomized trials	not serious	not serious	not serious	not serious	none	HIGH

* Other considerations: publication bias.

## Data Availability

All data are included in the manuscript.

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
