# Peer review of "Efficacy of Interventional Programs in Reducing Acculturative Stress and Enhancing Adjustment of International Students to the New Host Educational Environment: A Systematic Review and Meta-Analysis"

_ijerph, 2021, doi:10.3390/ijerph18157765_

Round 1

Reviewer 1 Report

I have read with interest this paper. I believe that the paper is interesting. However, I have some concerns that are reported herein.

Abstract should be improve; according to PRISMA Guideline “Abstracts can be useful for screening by study type,  facilitating quick assessment of validity; enabling efficient perusal of electronic search results; clarifying to which patients and settings the results apply; providing readers and peer reviewers with explicit summaries of results; facilitating the pre-publication peer review process;  and increasing precision of computerised searches”

Introduction is poor; it should be extended and to include the question review.

Line 82: there are a mistake “CINHHL” is not correct.

In general, the Data collection process , data items , study risk of bias assessment, effect measures and synthesis methods are confused. It should be clarified for better understanding.

In the discussion, authors should be discuss any limitations of the evidence included in the review,  any limitations of the review processes used and  implications of the results for practice, policy, and future research.

Author Response

Dear reviewer,

We want to provide our extreme thanks and appreciation to you for handling and reviewing our manuscript and for the valuable comments and recommendations to improve the quality of our manuscript.  Our response to your comments is given in the attached file point by point, and in the revised manuscript the modifications are highlighted in red colour.

Yours Sincerely,

Reviewer 2 Report

Thank you for the opportunity to review your paper entitled “Efficacy of Interventional Programs in Reducing Acculturative Stress and Enhancing Adjustment of International Students to The New Host Educational Environment: A Systematic Review and Meta-analysis”

Abstract

The abstract does not present adequate content. It is not necessary to indicate previous research or specifically detail the databases consulted. For more information, you can consult the PRISMA 2020 recommendations. The number of articles found, and finally selected, is detailed, but nothing is indicated about the type of analysis (meta-analysis, software, or size effects). It is no necessary indicate” the heterogeneity (I2) = 0% at P=0.69”, generate text with the meaning of this data. And the same with line 31 page. 1 “P<0.001, I2= 2% at P=0.31]”

Introduction

Introduction too short.  It is necessary to expand with data on the subject, and not only indicate the existence of articles without describing their results.

Material and Methods

Update with the new PRISMA 2020 guide (see http://prisma-statement.org/prismastatement/Checklist.aspx ) for diagram flow and checklist

Databases and information sources: Pag. 2 line 82, do the authors means “CINHAL “ and not “CINHHL” ?

Not necessary to explain “Selection bias” pag. 4 line 126. Remove from lines 126 to 166, and from 179 to 229.

Data extraction and collection repeats information already explained with the PICO strategy.

Not clear what is “Qualitative literature review”?

Quantitative literature review

It is not necessary to explain in an article to explain in detail the different ways of performing the meta-analysis. Summary of the type of meta-analysis performed.

Results

Remove Table 2.

Do not repeat the information, all the data in the first paragraphs of the results are already indicated in the flow chart

The number of articles in appendix A does not correspond to the total number of articles found.

I suggest to those who review other articles for systematic review and meta-analysis, as well as the PRISMA recommendations.

Author Response

Dear reviewer,

We want to provide our extreme thanks and appreciation to you for handling and reviewing our manuscript and for the valuable comments and recommendations to improve the quality of our paper.  Our response to your comments is given in the attached file point by point, and in the revised manuscript, the modifications are highlighted in red colour.

Yours Sincerely, 

Round 2

Reviewer 1 Report

This paper has been improved. I suggest accepting in present form.

Reviewer 2 Report

Thanks for the indicated changes